# Single-Cell RNA-Seq Revealed the Gene Expression Pattern during the In Vitro Maturation of Donkey Oocytes

**DOI:** 10.3390/genes12101640

**Published:** 2021-10-19

**Authors:** Zhipeng Li, Xinhui Song, Shan Yin, Jiageng Yan, Peiru Lv, Huiquan Shan, Kuiqing Cui, Hongbo Liu, Qingyou Liu

**Affiliations:** 1Laboratory for Conservation and Utilization of Subtropical Agro-Bioresources, Guangxi Univesity, Nanning 530005, China; songxinhui@st.gxu.edu.cn (X.S.); xiaobeiandme@foxmail.com (J.Y.); huiquan_shan@163.com (H.S.); kqcui@gxu.edu.cn (K.C.); qyliu-gene@gxu.edu.cn (Q.L.); 2Henan Chuangyuan Biotechnology Co. Ltd.; Zhengzhou 451100, China; yinshan@chuangyuanbio.com (S.Y.); peiru1116@163.com (P.L.)

**Keywords:** single-cell RNA-seq, donkey, oocytes, in vitro maturation, gene expression

## Abstract

Donkeys are an important domesticated animal, providing labor, meat, milk, and medicinal materials for humans. However, the donkey population is continuously declining and even at risk of extinction. The application of modern animal production technology, such as oocyte in vitro maturation, is a promising method to improve the donkey population. In this study, we explore the gene expression patterns of donkey germinal vesicle (GV) and in vitro matured metaphase II (MII) oocytes using single cell RNA-seq of the candidate genes along with the regulatory mechanisms that affect donkey oocyte maturation. We identified a total of 24,164 oocyte genes of which 9073 were significant differentially expressed in the GV and MII oocytes. Further Gene Ontology (GO) and Kyoto Encyclopedia of Genes and Genomes (KEGG) analysis indicated that these genes were associated with the meiotic cell cycle, mitochondrion activity, and N-glycan biosynthesis, which might be the key genes and regulatory mechanisms affecting the maturation of donkey oocytes. Our study provides considerable understanding regarding the maturation of donkey oocytes and serves as a theoretical basis for improving the development of donkey oocytes, which could ultimately benefit the expansion of the donkey population and conservation of biodiversity and genetic resources.

## 1. Introduction

Donkey (*Equus asinus*) belongs to the Equidae family and has had a close association with humans throughout history, as the donkey has greatly contributed to human mobilization for long-distance travel, load-bearing, and agriculture. Despite having multiple uses, donkeys were displaced due to rapid industrialization and mechanization, and the donkey population significantly decreased over the last century. This has placed almost all donkey breeds at risk, especially in the wild, for instance the Asiatic wild ass and the African wild ass [1,2]. In China, people value donkeys for varied purposes. Currently, donkeys are mainly used for producing meat, milk, and Chinese traditional medicines. Donkey meat is considered to be one of the most delicious meats owning to the content of linoleic acid, which is much higher than that of pork and beef, and this is good for human health [3]. Donkey milk is used as food supplement for special population groups due to its effects on human health, including antimicrobial activity, immunomodulation, and hypo-allergenicity [4]. Donkey-hide gelatin has a high nutritional value and can be used for medicinal purposes [5]. 

Despite these significant roles, this species is still hampered as donkeys are primarily raised conventionally, and they are at the threat of extinction. The use of animal assisted reproductive technology (ART), such as in vitro fertilization (IVF) and embryo transfer, will help improve donkey farming efficiency. However, little is known about the application of ART in donkeys. In vitro maturation (IVM) of oocytes and IVF are the indispensable techniques of assisted reproductive technologies and have been widely used in animal biotechnology and reproduction. The problem is that, currently, the IVM rates of equidae animals remain at a low level, and no research has included IVF on donkeys [1]. The major impairment to the development of equine ART lies in the lack of optimized conditions for in vitro maturation (IVM) of oocytes, in vitro fertilization (IVF), and cultures [2]. Oocyte maturation and meiosis are essential for mammals to produce developmentally competent haploid gametes for successful fertilization and embryo development [6]. However, the quality and developmental ability of oocytes matured in vitro is lower than that of oocytes matured in vivo [6], and the underlying molecular mechanism is still not fully revealed, which further constrains the application of ART in donkeys. During the meiotic maturation of oocytes, from the germinal vesicle (GV) to metaphase stage of second meiosis (MII), effective and faithful execution of key molecules, and signaling pathways were required [7]. Therefore, revealing the gene expression pattern during the maturation of donkey oocytes will be helpful to improve the efficiency of in vitro culture of donkey oocytes.

Single-cell RNA sequencing (scRNA-seq) has an advantage over the other omics-techniques to access the transcriptome changes using a small amount of molecular materials [8] and has widely been used to identify better quality oocytes for maturation [9]. Recently, scRNA-seq was used to investigate the effects of vitrification on the mRNA transcriptome of bovine GV oocytes and their in vitro-derived MII oocytes [10,11]. Important genes and molecular pathways related to maturation and development competence have been identified using scRNA-seq in humans [12], pigs [13], sheep [14], bovine [15] and mice [16]. ScRNA-seq has been used to unravel the key molecular ligands involved in the chemical-induced acceleration of oocyte maturation [17]. Thus, scRNA-seq could be a powerful tool to disclose the transcriptome changes of donkey oocyte development and can also determine the important mRNAs and their key pathways. In the present study, we intend to explore the gene expression patterns of donkey GV and IVM MII oocytes and reveal the candidate genes and their regulatory mechanisms affecting the maturation of donkey oocytes. This study provides novel understandings about donkey oocyte maturation, which serves as a theoretical basis for improving oocyte development and ultimately benefits the expansion of the donkey population and the conservation of biodiversity and genetic resources.

## 2. Materials and Methods

### 2.1. Ethics Statements

The breed of donkey used in this study is the Chinese Biyang donkey. All animal experiments were approved by the Animal Experiments Ethical Review Committee of the Guangxi University, Nanning, China (Grant NO.:Gxu-2020-138).

### 2.2. Chemicals

All the chemicals and reagents were purchased from Sigma-Aldrich (Shanghai, China). The IVM media was filtered (0.22 μm of pore size) and equilibrated at 38.5 °C under 5% CO_2_ for 2–3 h before use.

### 2.3. Collection and In Vitro Maturation (IVM) of Cumulus-Oocyte-Complexes (COCs)

The ovaries were collected at local commercial abattoirs and transported to the laboratory in 25 °C physiological saline supplemented with 0.1% (*v/v*) penicillin/streptomycin within 2 h. Ovaries were washed three times with physiological saline to wipe off blood and impurities and dissected to release cumulus-oocyte-complexes (COCs). COCs were collected using a scraping procedure that has previously been described [18]. Briefly, on the surface of the ovary and within the ovarian stroma, all follicles between 5 and 25 mm in diameter were opened with a scalpel blade, and the granulosa cell layer was scraped with a curette. 

COCs were recovered from the collected mural granulosa cells using a dissection microscope. All rinsed fluids were examined for oocyte recovery under an inverted optical microscope. Denuded and degenerated oocytes showing shrunken, dense, or fragmented cytoplasm, were discarded. COCs were then transferred into four-well plates with 1 mL maturation medium per well under oil and cultured for 34 h at 38.5 °C in humidified air (100% humidity) containing 5% CO_2_. The maturation medium consisted of TCM-199, supplemented with 10% (*v/v*) fetal calf serum (FCS), 50 ng/mL epidermal growth factor, 100 ng/mL insulin-like growth factor 1, 0.1 IU/mL porcine FSH, and 0.1 IU/mL porcine LH.

### 2.4. Collection of Donkey Oocytes

The immature GV oocytes and in vitro mature MII oocytes were used for RNA-seq. TCM199 with 10% FCS containing 0.1% hyaluronidase was used to remove the cumulus cells and granular cells of the GV and in vitro matured MII oocytes. MII oocytes were confirmed with the presence of the first polar body near the zona pellucida. They were selected and washed in DPBS three times and transferred into lysis solution at −80 °C until further study. Each sample contained 10 oocytes, and each group was repeated three times.

### 2.5. Library Preparation and RNA Sequencing

All sequencing work was performed by the Annoroad Gene Technology Co.; Ltd.; (Beijing, China). Smart-Seq2 method was used for first-strand cDNA synthesis to establish a sequencing library. Briefly, mRNAs were first segmented in a fragmentation buffer, followed by first-strand cDNA synthesis. The cDNA was broken into small fragments of about 350 bp, followed by end repair, poly-A tail addition, and sequencing adapter addition. Library purification was accomplished by Beckman Ampure XP beads, and the final sequencing library was obtained by PCR. In total, the library was sequenced using Illumina NovaSeq 6000, which produced paired-end libraries with a 150 bp (PE150) read length.

### 2.6. Identification of Differentially Expressed Gene

The clean reads were uploaded in the NCBI Sequence Read Archive (Accession number PRJNA763991) and were processed with BMK Cloud (www.biocloud.net, accessed on 21 January 2021; Biomarker Technologies Co. Ltd.; Beijing, China). Briefly, the clean reads were mapped to the donkey (*Equus_asinus*) genome (ASM130575v1). The expression level of the gene was estimated according to fragments per kilobase of exon per million fragments mapped (FPKM) values using Cufflinks software. 

DESeq and *p*-values were used to evaluate the differential gene expression between GV and MII oocytes. Gene abundance differences between samples were then calculated based on the ratio of the FPKM values. The false discovery rate (FDR) control method was used to identify the threshold of the *p*-value in multiple tests to calculate the significant differences. The FDR < 0.05 and |log_2_ (fold change) |≥ 1.5 values were used as a criterion to find the differentially expressed genes (DEGs), which were further used for subsequent analysis.

### 2.7. Gene Ontology (GO) and Kyoto Encyclopedia of Genes and Genomes (KEGG) Analyses

GO and KEGG analysis of the differentially expressed genes (http://www.geneontology.org/ and http://www.enome.jp/kegg/; accessed on 22 January 2021) was capable of discovering the key candidate genes in the regulatory network on the basis of molecular pathways and biological processes, and the criteria for significance were set at a *p*-value of < 0.05.

### 2.8. qRT-PCR Analysis

The immature GV oocytes from donkey ovaries and in vitro matured MII oocytes were collected with 8 µL cell lysis buffer (cells-to-cDNA^TM^ II kit, Ambion, Shanghai, China). The total RNA was extracted and reverse transcribed to cDNA using the SuperScript^TM^ II Reverse Transcriptase Kit (Invitrogen, Shanghai, China) according to the manufacturer’s instructions. In brief, the 20 µL mixture of reverse transcription system contained 8 µL of cell lysis buffer with six oocytes, 1 µL of DNase, 1.3 µL of DNase I buffer, 1 µL of EDTA, 1 µL of dNTP, 2 µL of random primer, 4 µL of 5 × first-stand buffer, 0.5 µL of RNase inhibitor, 2 µL of DTT, and 0.25 µL of FS RT (reverse transcriptase) enzyme. 

Before quantitative real-time polymerase chain reaction (qRT-PCR) amplification, the primers of reference gene *ACTB* and target genes (*GDF9*, *BMP15*, *LGALS3*, and *ALG5*) were designed using Oligo 7.0 software (Appendix A), and all primer synthesis work was done by GenSys Biotech (Nanning, China). The relative expression of all genes was quantified using the SYBR qPCR Master Mix (Vazyme, Shanghai, China) on the basis of the manufacturer’s instructions. Fluorescence data were acquired using a fluorescence ratio PCR instrument (Roche, Shanghai, China). Finally, the relative gene expression was calculated using the 2^−^^△△Ct^ method.

### 2.9. Statistical Analysis

All experiments were repeated three times, and SPSS version 17.0 was used for data analyses. Differences between the groups were analyzed with a *t*-test. Values are expressed as the mean ± SEM, and a value of *p* < 0.05 was used to indicate statistical significance.

## 3. Results

### 3.1. In Vitro Development of Donkey Oocytes

We collected a total of 1692 COCs from 229 donkey ovaries, with an average of 7.33 COCs per ovaries (Table 1). For in vitro maturation, five repetitions were performed to evaluate the maturation rate. As shown in Table 2, about 1591 COCs were used for IVM, and 423 MII oocytes with first polar body extrusion (27.08 ± 3.23%) were obtained after maturation. The rate of MII oocytes was significantly higher (*p* < 0.01) than that of the MI oocytes (9.26 ± 1.18%). These results suggested that a large number of oocytes could be harvested from the ovaries, but most of them were stunted at the GV stage.

### 3.2. scRNA-seq of Donkey Oocytes

GV and MII oocytes with uniform cytoplasm were collected (Figure 1A), and scRNA-seq was performed for the gene expression pattern analysis. Principal component analysis (PCA) showed that GV and MII oocytes were clustered into two groups (Figure 1B). The Pearson correlation coefficient between oocytes for replicates was above 0.95 (ranging from 0.950 to 0.974), which was in accordance with the recommendations of the ideal sampling and test conditions of the Human Genome Project [19] (Figure 1C). 

In the present study, 21,055, 21,502, and 21,736 transcripts (FPKM > 0) were identified in the GV oocytes, and 20,043, 20,141, and 20,513 mRNAs were identified in the MII oocytes (Figure 1D), suggesting that the number of maternal RNAs decreased gradually with oocyte meiosis maturation. In total, 24,164 genes were identified in the oocytes; 1719 and 882 genes were specifically expressed in the GV and MII oocytes, respectively (Figure 1E).

### 3.3. Abundant mRNAs in Oocyte Meiosis

Further, we analyzed those abundantly expressed genes, which were considered to be vital to maintain the basic characteristics of maturation. The top 20 mRNAs in each group were selected, and nine of them, including *DPPA3*, *PTTG1*, *BTG4*, *KPNA7*, *RNF34*, *UBB*, *ZP3*, *CNBP*, and *GDF9*, were annotated (Table 3). Furthermore, *AGR2*, *ZAR1L*, *ARG2*, *BMP15*, *CALM2*, *RFC3*, *LOC106840558*, and a novel gene (*NewGene_26554*), were found to be highly abundant in GV oocytes. While *RASL11A*, *WEE2*, *CNBP*, *CCNB1*, *CENPE*, *HNRNPA1*, *LOC106829384*, and *NewGene_15553* were found in the MII group.

### 3.4. Differentially Expressed Genes (DEGs) in GV and MII Oocytes

To better understand the molecular events during in vitro maturation of donkey oocytes, the differentially expressed genes between GV and MII, oocytes were analyzed using DESeq2 (Figure 2A). In total, 9073 genes were significantly differentially expressed (FDR < 0.05 and |log_2_ (fold change) | ≥ 1.5). Among them, 4388 genes were significantly highly expressed in the MII oocytes, while 4685 genes were highly expressed in the GV oocytes (Figure 2B,C). The top 10 annotated DEGs (up- or down- regulated) were shown in Table 4 (ordering from the largest|fold change |). This data suggested that genes relation to cell cycle and mitochondrion activity may be key candidate genes affecting donkey oocyte maturation.

### 3.5. GO and KEGG Analyses of DEGs in GV and MII Oocytes

GO and KEGG enrichment analysis were performed to further understand the function of the DEGs. GO analysis of the up-regulated genes showed that they were mainly enriched in biological processes, including DNA endoreduplication, DNA repair, RNA binding, and protein N- and C- terminus binding (Figure 3A). The down-regulated genes were mainly enriched in biological processes related to mitochondrial activity, such as the mitochondrion, mitochondrial matrix, mitochondrial inner membrane, and NADH dehydrogenase (ubiquinone) activity, indicating that the process of oocyte maturation is accompanied by changes in mitochondrial activity (Figure 3B). 

Furthermore, GO terms were found in relation to the transcriptional and translation states, including rRNA processing, RNA binding, rRNA binding, cytosolic large/small ribosomal subunit, formation of translation preinitiation complex. Thus, from the above stated results, we found that genes involved in the meiotic cell cycle were highly expressed in MII oocytes, while genes that contributed to transcription and mitochondrial activity were highly expressed in GV oocytes.

Moreover, KEGG analysis showed that the up-regulated genes were enriched in 291 signaling pathways, including the cell cycle, RNA transport/degradation, ubiquitin mediated proteolysis, N-Glycan biosynthesis, basal transcription factors, adherens junction, oocyte meiosis, DNA replication, and lysosome, which may affect the oocyte fate during the transition from GV to MII (Figure 4A). KEGG analysis of down-regulated genes found that these were mainly enriched in the pathways related to ribosomes, non-alcoholic fatty liver disease (NAFLD), the spliceosome, the proteasome, RNA polymerase, the pyrimidine metabolism, the pyruvate metabolism, oxidative phosphorylation, and the citrate cycle (Figure 4B).

### 3.6. Glycosylation Vital for Oocyte In Vitro Maturation

Among the pathways enriched by up-regulated genes, we noticed that the N-Glycan biosynthesis pathway was significantly enriched, while it was not present in other mammalian oocytes. Further analysis showed that this pathway was associated with 20 DEGs, including *DPAGT1*, *ALG5*, *MGAT5*, *MAN1A1*, etc. (Figure 5A). Gene relations to the mucin type O-Glycan biosynthesis pathway were hierarchically clustered and are presented in Figure 5B. These mainly belong to the UDP-N-acetyl-α-D-galactosamine:polypeptide N-acetylgalactosaminyltransferase (GalNAc-T) and β-1,6-N-acetylglucosaminyltrans-ferase gene family. The high expression of glycosylation genes during oocyte maturation suggests that protein N-glycosylation and O-glycan biosynthesis in oocytes might have a crucial role in oocyte development in donkeys.

### 3.7. qPCR Validation

To confirm the results of the scRNA-seq analysis, the expression of four DEGs (two down-regulated and two up-regulated) were tested using qRT-PCR. The results showed that the expression of *GDF9* and *BMP15* were significantly down-regulated (*p* < 0.001), while *LGALS3* and *ALG5* were significantly up-regulated (*p* < 0.05) (Figure 6). This result is consistent with the scRNA-seq data above.

## 4. Discussion

Mammalian oocyte maturation is a coordinated process under the control of multiple biological events and is critically regulated by a molecular network mainly composed of maternal mRNAs deposited in oocytes [6]. Oocyte maturation has been widely studied in pigs, cattle, and buffalo [20,21]; however, very little is known about the donkey species. In this study, we analyzed the transcriptomes of donkey GV and MII oocytes using scRNA-seq, and identified 1719 and 882 specifically expressed genes in the GV and MII oocytes, respectively. Further DEGs, GO, and KEGG analysis showed that these genes were found to be associated with the meiotic cell cycle, mitochondrion activity, and N-glycan biosynthesis, which may be the key mechanisms influencing donkey oocyte maturation.

Like other mammals, we also observed that the total transcripts were decreased during the IVM of donkey oocytes. The possible reason is the transcriptional silencing caused by chromatin condense after nuclear membrane breakdown [22] and that deposited maternal RNAs were selectively degraded during the GV to MII transition [23]. Maternal genes play important roles in oocyte maturation, and *GDF9* and *BMP15,* which are considered essential during oocyte development, maturation, and fertilization, were also identified in a comparison of bovine GV and MII oocytes [15,24,25,26,27]. Down-regulating *GDF9* can induce intraovarian hyperandrogenism and finally lead to follicular arrest [28]. A lower quality of oocytes in patients with a poor ovarian response correlated with a decreased expression of *GDF9* and *BMP15* in follicular fluid (FF) and granulosa cells (GCs) [29]. In this study, the expression of maternal genes was analyzed using RNA-seq, and the expression of *GDF9* and *BMP15* was further tested by qRT-PCR. We found that the maternal genes, including *GDF9, BMP15, ZAR1, NLRP5, PADI6,* and *NPM2,* were significantly down-upregulated during in vitro maturation, suggesting that the down-regulation of these genes may be some of the reason for the low development efficiency of donkey oocytes. We also found multiple highly expressed genes during oocyte maturation, suggesting that these genes may play indispensable roles. For example, the expression of *PTTG1* was found to be maintained at a high level in GV- and MII-stage oocytes, and was considered to participate in the processes of oocyte maturation and zygotic genome activation during porcine embryogenesis [30]. The *PTTG1* was also found to be abundant in GV and MII donkey oocytes in our study, which is consistent with the reported results. Moreover, we also found several novel genes (NewGene_15562, NewGene_15553, and NewGene_26555) that may exert vital functions in donkey oocyte development.

We further analyzed the DEGs between the GV and MII oocytes and identified 4388 and 4685 significant high expression genes in GV and MII oocytes, respectively. Among the top 10 up-regulated genes, *LGALS3* is the most significant one, and earlier it was reported that *LGALS3* can mediate human spermatozoa-zona pellucida binding and affect the in vitro fertilization process [31]. Thus, we hypothesized that the increased expression of *LGALS3* is important to prepare for in vitro fertilization. Furthermore, multiple up-regulated genes have been found to be associated with fertilization, including *TFPI2* [32] and *S100A11* [33], suggesting that the acquisition of the fertilization ability of oocytes is an important event of oocyte maturation. *ANXA1* and *AKAP2* were also highly expressed in in vitro mature MII oocytes of bovine and mice [34,35]. Cyclin D2 (*CCND2*) is responsible for ovarian cell proliferation, and for regulating the G1/S phase transition during the cell cycle [36]. For the top significant decreased genes, evidences showed that *ATP5J2*, *SMDT1*, and *NDUFA8* were associated with mitochondrial activity [37,38,39], which suggests that oocytes might rely on oxidative phosphorylation in the mitochondrion to satisfy the energy requirements of oocyte maturation. It has been suggested that 87% of ATP production in the equine COC is from oxidative phosphorylation [40]. Through GO and KEGG analysis of the DEGs, we found that the downregulated pathways were related to the energy metabolism, including the TCA cycle, pyruvate metabolism, and oxidative phosphorylation. These pathways were also enriched during the GV to MII transition of porcine oocytes [13]. ATP is produced via the TCA cycle and oxidative phosphorylation (OXPHOS) in mitochondria, which derive from oxidizing pyruvate in combination with endogenous fuel sources [41]. The addition of pyruvate in the maturation medium may increase the energy supply to a certain extent, thereby, increasing the maturation rate of oocytes. Additionally, the down-regulated genes were also enriched in glycolysis/gluconeogenesis, which was also found in a study on bovine oocytes [15]. However, this does not indicate a relatively low efficiency of glucose utilization in MII oocytes compared to GV oocytes. Li et al. compared the metabolic patterns of mice GV, MI, and MII oocytes and found that carbohydrates were increased during mouse oocyte meiotic resumption and meiotic maturation [42]. 

We further analyzed the N-glycan biosynthesis pathway and found 20 DEGs, including the *MGAT5, DPAGT1,* and *MAN1A1*. *MGAT5* and *DPAGT1* are key enzymes catalyzing protein N-glycosylation [43]. Studies have shown that the silencing of *MGAT5* resulted in the decrease of Galectin-3 binding [44]. Missense mutations (c.497A >G; *p*. Asp166Gly) or the knockout of *DPAGT1* reduced the glycosylation of ZP proteins causing infertility in female mice [45]. *MAN1A1* codes for mannosidase A, which hydrolyses terminal mannose residues. In cattle, removing D-mannosyl residues from the zona pellucida was found to be interruptive for oocyte–sperm binding [46]. In this study, the genes relation to N-glycan-related were significantly up-regulated during IVM, suggesting that N-glycosylation was one of the key factors affected the in vitro maturation of donkey oocytes.

In this study, as we did not perform an oocyte viability test, the poor maturation rate of oocytes may have been caused by the poor viability of GV oocytes collected from the ovaries. In addition, due to the limitations of experimental materials, we only studied the mRNA transcription of the oocytes. In the oocytes, the expression patterns of RNA and proteins are not always correlated. Down-regulated mRNA may even produce more abundant proteins. Moreover, it is better to use the in vivo matured oocytes as a control group in a study. However, we did not obtain the farmer’s consent for this, as the donkey embryos are too precious. In this study, we selected the best quality MII oocytes for RNA-seq as much as possible according to the embryo morphology evaluation. Considering that these oocytes can develop into embryos as the in vivo matured oocytes do, we propose that their transcripts are similar to a certain extent. In addition, our study mainly focused on the gene expression patterns of in vitro matured donkey oocytes, and, while artifacts during in vitro culture may change the transcriptional profiles to an extent when comparing the in vivo maturated oocytes, the data and analysis about in vitro matured donkey oocytes are representable.

Oocyte maturation in vitro is the basis of in vitro embryo production. Improving the oocyte in vitro maturation rate can promote the research and application of assisted reproduction technology (ART). Assisted reproductive technologies, including in vitro fertilization (IVF), somatic cell nuclear transfer (SCNT), and embryo transfer (ET), have been widely used in animal genetic improvement and population expansion, such as the breeding of cows, pigs and other animals, and the population conservation and expansion of pandas. Donkeys are also an important part of human society and of biodiversity. Improving the in vitro maturation rate of donkey oocytes can provide a basis for the application of ART in donkeys, which would ultimately benefit the expansion of the donkey population and conservation of biodiversity and genetic resources.

## 5. Conclusions

In conclusion, we obtained the transcriptome patterns during donkey oocyte maturation using scRNA-seq, and identified numerous DEGs that were crucially involved in oocyte maturation. Further analysis found that the genes related to the meiotic cell cycle, mitochondrion activity, and N-glycan biosynthesis might be candidate genes that regulate the key regulatory mechanisms affecting the maturation of donkey oocytes. This study provides considerable understanding regarding the maturation of donkey oocytes, and also serve as a theoretical basis for improving the development of donkey oocytes, with the intention to ultimately benefit the expansion of the donkey population and conservation of biodiversity and genetic resources.

## Figures and Tables

**Figure 1 genes-12-01640-f001:**
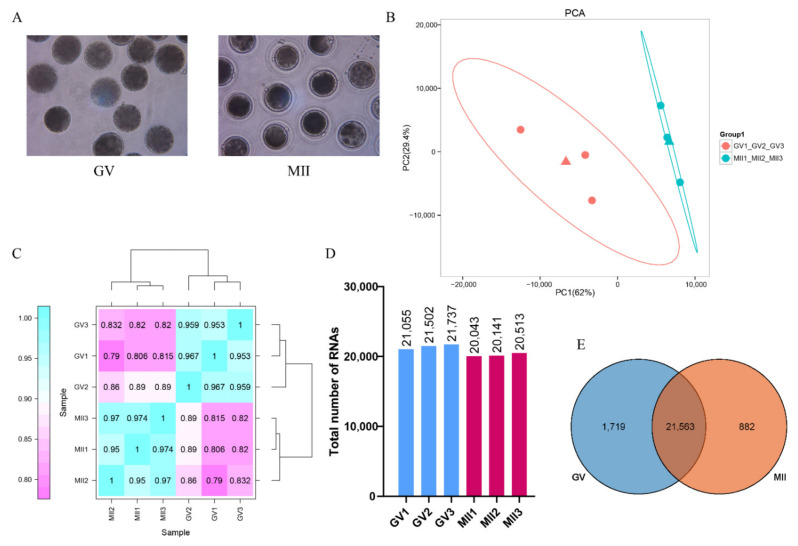
Transcriptional analysis of donkey oocytes at the GV and MII stage. (**A**). Collected donkey oocytes at the GV and MII stage for scRNA-seq. (**B**). Principal component analysis (PCA) of six samples based on gene expression. The red dots indicate GV oocytes, and the blue dots indicate MII oocytes. (**C**). Repeated relevance assessment of the transcriptomes of donkey oocytes. (**D**). The total mRNAs detected by scRNA-seq in GV and MII oocytes. (**E**). Overlap of the expressed genes identified in GV and MII oocyte.

**Figure 2 genes-12-01640-f002:**
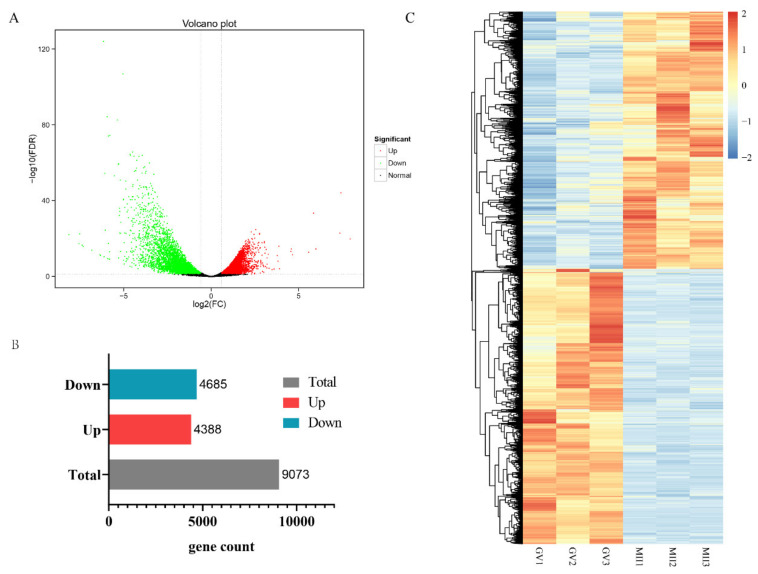
Differentially expressed gene (DEG) analysis. (**A**). Volcano plot for DEGs between GV and MII donkey oocytes (FDR < 0.05 and |log_2_ (fold change) | ≥ 1.5); The red dots represent upregulated genes, and the blue dots represent downregulated genes. (**B**). Bar chart showing the up-regulated DEGs and down-regulated DEGs. (**C**). Heatmap of all DEGs.

**Figure 3 genes-12-01640-f003:**
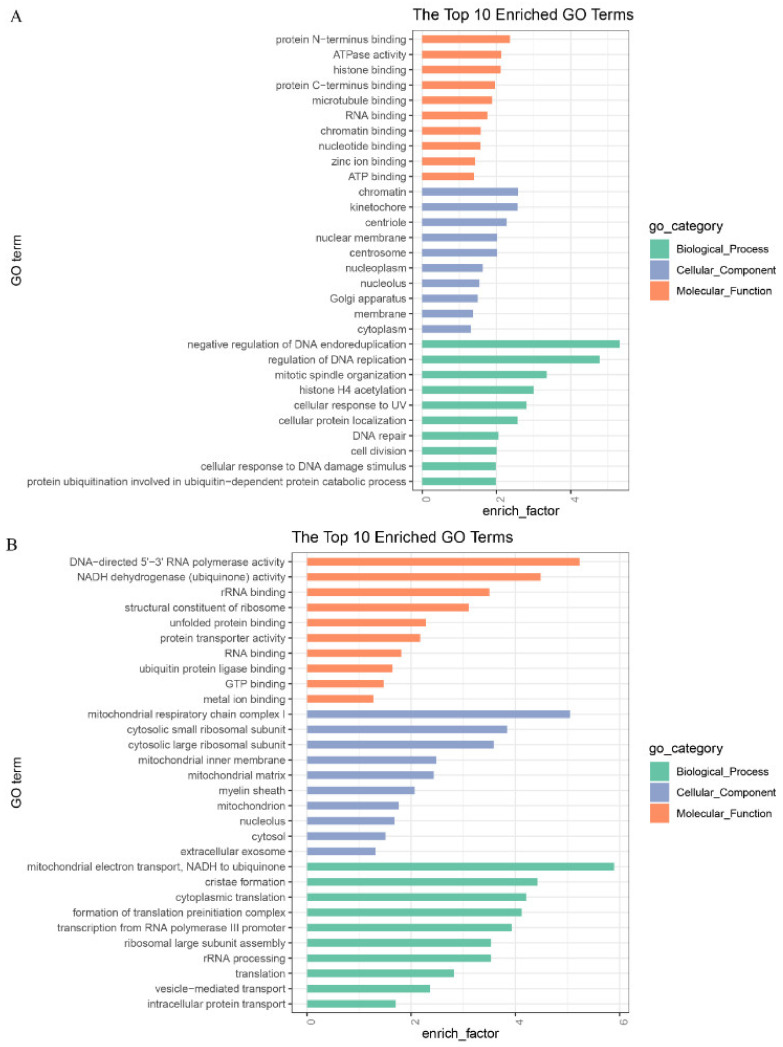
GO enrichment of (**A**) up-regulated DEGs and (**B**) down-regulated DEGs with the top 10 terms in biological process, cellular component, and molecular function.

**Figure 4 genes-12-01640-f004:**
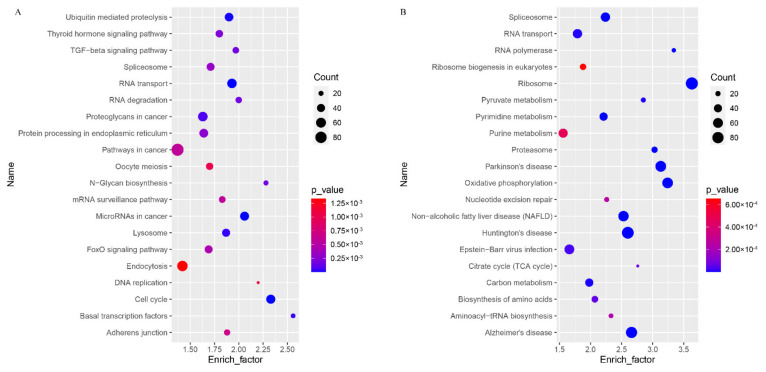
KEGG enrichment of the top 20 (**A**) up-regulated and (**B**) down-regulated terms.

**Figure 5 genes-12-01640-f005:**
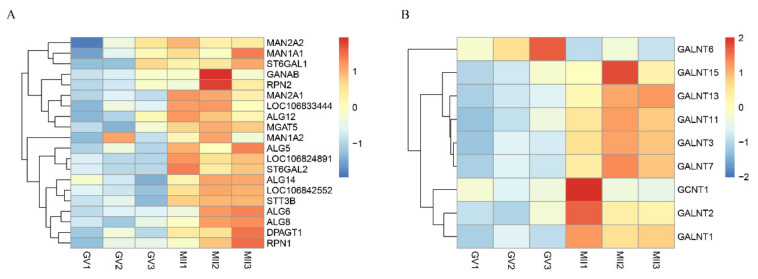
Expression of glycosylation genes. (**A**). Heat map representation of DEGs belonging to the selected N-Glycan biosynthesis enriched by up-regulated DEGs. (**B**). Heat map representation of DEGs belonging to the selected mucin-type O-Glycan biosynthesis enriched by all DEGs.

**Figure 6 genes-12-01640-f006:**
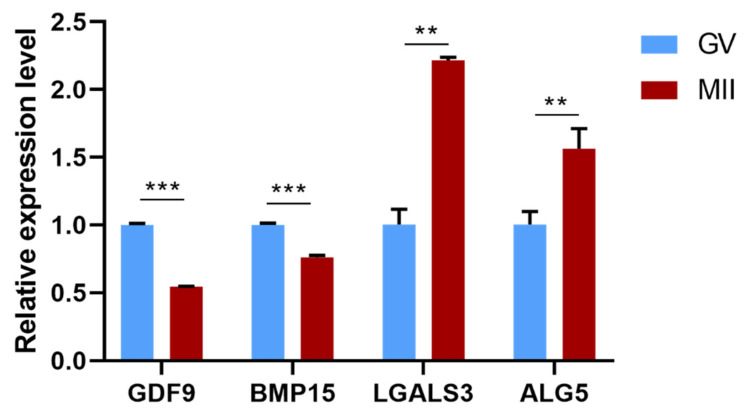
The expression of differentially expressed genes during oocyte maturation. ** indicates *p* < 0.01; and *** indicates *p* < 0.001.

**Table 1 genes-12-01640-t001:** Oocyte recovery from adult jenny ovaries by scraping.

Repetitions	No. of Ovaries Per Experiment	No. of Oocytes Recovered	No. of Oocytes Per Ovaries
1	48	361	7.52
2	45	291	6.47
3	48	365	7.60
4	55	445	8.09
5	33	230	6.96
Total	229	1692	7.33 ± 0.28 *

Note: * The data are mean ± SEM from five replicates.

**Table 2 genes-12-01640-t002:** Nuclear maturation of donkey oocytes after in vitro culture.

Oocytes Stage	No. of Oocytes (%)
GV	1018 (63.68 ± 2.68 ^c^)
MI	150 (9.26 ± 1.18 ^a^)
MII	423 (27.08 ± 3.23 ^b^)

Note: All data are the mean ± SEM from five replicates. Different superscripts within the same column represent significant differences (*p* < 0.01).

**Table 3 genes-12-01640-t003:** The top 20 most abundantly expressed genes in the scRNA-seq profiles of each group.

GV Group (*n* = 3)			MII Group (*n* = 3)		
Gene Name	Gene Description	Average FPKM	Gene Name	Gene Description	Average FPKM
*DPPA3*	Developmental Pluripotency-Associated Protein 3	20,656.8	*DPPA3*	Developmental Pluripotency-Associated Protein 3	20,223.6
*PTTG1*	Pituitary Tumor-Transforming 1	19,649.5	*PTTG1*	Pituitary Tumor-Transforming 1	19,585.9
*LOC106824246*		10,616.3	*NewGene_15562*		7462.3
*GDF9*	Growth Differentiation Factor 9	6493.5	*BTG4*	BTG Anti-Proliferation Factor 4	5170.7
*UBB*	Ubiquitin B	6060.0	*LOC106824246*		5137.5
*AGR2*	Anterior Gradient 2, Protein Disulphide Isomerase Family Member	5435.3	*RASL11A*	RAS Like Family 11 Member A	4670.0
*NewGene_15562*		4559.9	*KPNA7*	Karyopherin Subunit α 7	4645.6
*ZP3*	Zona Pellucida Glycoprotein 3	4516.6	*CKS1B*	CDC28 Protein Kinase Regulatory Subunit 1B	4343.5
*CKS1B*	CDC28 Protein Kinase Regulatory Subunit 1B	4381.1	*NewGene_15553*		3948.4
*RNF34*	Ring Finger Protein 34	3478.1	*RNF34*	Ring Finger Protein 34	3662.4
*BTG4*	BTG Anti-Proliferation Factor 4	3358.6	*UBB*	Ubiquitin B	3030.3
*ZAR1L*	Zygote Arrest 1 Like	3323.5	*ZP3*	Zona Pellucida Glycoprotein 3	2997.1
*ARG2*	Arginase 2	3278.5	*WEE2*	WEE2 Oocyte Meiosis Inhibiting Kinase	2327.4
*KPNA7*	Karyopherin Subunit α 7	3226.7	*LOC106829384*		2294.0
*LOC106840558*		2823.5	*CNBP*	CCHC-Type Zinc Finger Nucleic Acid Binding Protein	2293.9
*NewGene_26555*		2684.6	*CCNB1*	Cyclin B1	2198.8
*LOC106844365*		2541.2	*CENPE*	Centromere Protein E	2163.2
*BMP15*	Bone Morphogenetic Protein 15	2472.0	*GDF9*	Growth Differentiation Factor 9	2068.7
*CALM2*	Calmodulin 2	1989.0	*HNRNPA1*	Heterogeneous Nuclear Ribonucleoprotein A1	2042.1
*RFC3*	Replication Factor C Subunit 3	1935.0	*LOC106844365*		1945.9

**Table 4 genes-12-01640-t004:** Top 10 genes with the highest and the lowest fold changes after oocyte in vitro maturation.

Gene Symbol	Gene Name	Fold Change	*p*-Value
*LGALS3*	Galectin-3	7.94	1.42 × 10^−20^
*TFPI2*	Tissue Factor Pathway Inhibitor 2	7.42	7.93 × 10^−45^
*ANXA1*	Annexin A1	7.35	1.54 × 10^−23^
*S100A11*	S100 Calcium Binding Protein A11	5.99	2.95 × 10^−15^
*AKAP2*	A-Kinase Anchoring Protein 2	5.86	4.08 × 10^−34^
*ACTG2*	Actin γ 2, Smooth Muscle	5.57	1.41 × 10^−13^
*INHBA*	Inhibin Subunit β A	4.57	3.00 × 10^−15^
*LOC106825888*		4.56	5.77 × 10^−14^
*VIM*	Vimentin	3.91	8.33 × 10^−5^
*CCND2*	Cyclin D2	3.87	8.85 × 10^−17^
*ATP5J2*	ATP Synthase Membrane Subunit F	−7.52	3.40 × 10^−23^
*LOC106837922*		−7.32	1.52 × 10^−16^
*FKBP2*	FKBP Prolyl Isomerase 2	−6.37	3.40 × 10^−12^
*SMDT1*	Single-Pass Membrane Protein With Aspartate Rich Tail 1	−6.33	7.50 × 10^−12^
*TUBB4B*	Tubulin β 4B Class IVb	−6.15	8.38 × 10^−125^
*LOC106847707*		−6.13	4.16 × 10^−11^
*LOC106823315*		−6.08	3.44 × 10^−55^
*TMEM11*	Transmembrane Protein 11	−6.06	4.10 × 10^−25^
*NDUFA8*	NADH: Ubiquinone Oxidoreductase Subunit A8	−6.06	4.10 × 10^−25^
*LOC106836642*		−5.95	4.85 × 10^−85^

## Data Availability

The data that support the findings of this study are available from the corresponding author upon reasonable request.

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
