# Peer review of "Single-Cell RNA-Seq Revealed the Gene Expression Pattern during the In Vitro Maturation of Donkey Oocytes"

_genes, 2021, doi:10.3390/genes12101640_

Round 1

Reviewer 1 Report

The manuscript “Single-Cell RNA-seq Revealed Candidate Genes Affect Don-2 key Oocytes in Vitro Maturation” by  Zhipeng Li et al reported the transcriptomic study which compared matured and non-matured donkey oocytes. Although similar studies were performed in other species of farms animals, the work is solid and reported the novel data.  However, the manuscript requires several corrections and additional information.

  1. At the beginning of the introduction, please indicate the species of the studies that you have cited.
  2. The main question is : were GV oocytes, which were used for RNAseq, taken before or after IVM? In other words, GV oocytes were from the ovaries, or taken from 1018 oocytes, which were not matured after 34h of IVM (Line 95)?
  3. L101 Please refine the method of oocyte collection. Do you mean that the COCs after IVM were put in 0.1% hyaluronidase-containing medium, and after stripping the GV and MII oocytes were selected? Or, immature GV oocytes for RNAseq were recovered from the COCs before IVM culture?
  4. Was the test on oocyte viability performed after IVM? Because most of the oocytes did not progress to Metaphase-II during IVM, the question about the viability of GV oocytes can be raised.
  5. The values of three replicates in the Table 3 could be placed to supplementary data; the list of the top 20 the most abundant transcripts can be given with only the mean expression values; also the gene names (description) and not only gene symbols should be shown.
  6. Figure 3 is not readable and could be divided in two figures (GO enrichment and KEGG path enrichment )
  7. The text and legends should be extensively corrected for English and type errors
  8. Discussion sometimes appears as a catalogue of genes with their known functions; however, RNA and protein patterns are not correlated in the oocytes, and down-regulated mRNA may give more abundant protein and vice versa. You should mention this.
  9. It will be interesting to compare GO enrichment patterns between GV and MII oocytes discovered in this study and in other species, and explain the similarities and differences.
  10. What about the abundance of maternal transcripts, because the huge amount of RNAs are degraded during meiosis in mammals? Please develop more this in the discussion
  11. The perspective to improve poor oocyte maturation rate of donkey oocytes in vitro may be also discussed.

Author Response

Thank you very much for your suggestions and the comments, we have revised our manuscripts point by point according to these comments and suggestions. We wish that our revised manuscript will be qualified enough to be accepted. If there is any questions, please don't hesitate to contact me.

Response to Reviewer 1:

  1. At the beginning of the introduction, please indicate the species of the studies that you have cited.

Reply: Thank you for your suggestion. However, I’m sorry that we didn’t get your words. At the beginning of the introduction, we are trying to illustrate the important role of donkeys in human history. We're talking about the donkey, not any specific donkey species. The donkeys used in this study is China Biyang Donkey (please see in line 84), which have been point out in the method section. Hope you can provide more detail description of this question so that we can give a more accurate answer.

  1. The main question is: were GV oocytes, which were used for RNAseq, taken before or after IVM? In other words, GV oocytes were from the ovaries, or taken from 1018 oocytes, which were not matured after 34h of IVM (Line 95)?

Reply: Thank you for your question. In this study, the GV oocytes used for RNA-seq were taken from the ovaries without IVM. We have supplemented the detail description of the GV oocytes in the paper (please see in line 109).

  1. L101 Please refine the method of oocyte collection. Do you mean that the COCs after IVM were put in 0.1% hyaluronidase-containing medium, and after stripping the GV and MII oocytes were selected? Or, immature GV oocytes for RNAseq were recovered from the COCs before IVM culture?

Reply: Thank you for your reminder. In this study, the GV oocytes used for RNA-seq were immature GV oocytes recovered from the COCs before IVM. The 0.1% hyaluronidase-containing medium was used for stripping the cumulus cells and granulosa cells around the immature GV and IVM MII oocytes before RNAseq. We have revised the method of oocyte collection in the paper (please see in line 110).

  1. Was the test on oocyte viability performed after IVM? Because most of the oocytes did not progress to Metaphase-II during IVM, the question about the viability of GV oocytes can be raised.

Reply: This is a good suggestion that we have omitted. I agree with your speculation that the poor viability of GV oocytes may led to the poor maturation rate of oocytes. However, I'm sorry that we didn’t perform the oocyte viability test in this study. This is the first time for our team conducting research on donkey oocytes, and we will supplement this test in future studies. We also mentioned this defect in the discussion (please see in line 345).

  1. The values of three replicates in the Table 3 could be placed to supplementary data; the list of the top 20 the most abundant transcripts can be given with only the mean expression values; also the gene names (description) and not only gene symbols should be shown.

Reply: Thank you for your helpful suggestions. We have revised Table 3 according to the suggestions. The list of the top 20 most abundant transcripts was given with only the mean expression values and the replicate data were moved to supplementary Table S2. The gene names were also supplemented in Table 3 (please see in line 202).

  1. Figure 3 is not readable and could be divided in two figures (GO enrichment and KEGG path enrichment)

Reply: Thank you for your helpful suggestions. We have divided Figure 3 into two figures (Figure 3 and Figure 4) to better presentation in the paper (please see in line 242 and 245).

  1. The text and legends should be extensively corrected for English and type errors

Reply: Thank you for your suggestion. We have invited a friend who is native English speaker to improve and revise the writing of this paper (please see in line 395).

  1. Discussion sometimes appears as a catalogue of genes with their known functions; however, RNA and protein patterns are not correlated in the oocytes, and down-regulated mRNA may give more abundant protein and vice versa. You should mention this.

Reply: Thank you for your question. In this study, Due to the limitation of experimental materials, we only studied the mRNA transcription of the oocytes. Studies on protein expression may further reveal the influencing factors of oocyte maturation in vitro. We have supplemented and revised the discussion about this in the discussion. (please see in line 347).

  1. It will be interesting to compare GO enrichment patterns between GV and MII oocytes discovered in this study and in other species, and explain the similarities and differences.

Reply: Thank you for your suggestion. We have supplemented the comparison of GO and KEGG enrichment results with other species and discussed the similarities and differences between them (see details in line 324 and 330). For example, the maternal genes GDF9 and BMP15 were also identified during bovine IVM oocytes, suggesting the important roles of GDF9 and BMP15 in oocytes maturation (see details in line 287).

  1. What about the abundance of maternal transcripts, because the huge amount of RNAs are degraded during meiosis in mammals? Please develop more this in the discussion

Reply: Thank you for your question. Although the total amount of maternal transcripts is abundant, the expression of most maternal genes is reduced during in vitro maturation based on the RNA-seq data. The expression of two maternal genes, GDF9 and BMP15, were further analyzed using qRT-PCR and results showed that they were down-regulated during the IVM, which consist with the RNA-seq results. We have added this result in the paper (see details in line 285).

  1. The perspective to improve poor oocyte maturation rate of donkey oocytes in vitro may be also discussed.

Reply: Thank you for your suggestion. We have supplemented the perspective of improving donkey oocytes in vitro maturation in the end of the discussion as follows. Oocyte maturation in vitro is the basis of in vitro embryo production. Improving oo-cyte in vitro maturation rate can promote the research and application of assisted repro-duction technology (ART). Assisted reproductive technologies, including in vitro fertiliza-tion (IVF), somatic cell nuclear transfer (SCNT) and embryo transfer (ET), have been wide-ly used in animal genetic improvement and population expansion, such as the breeding of cows, pigs and other animals, and population conservation and expansion of pandas. Donkeys are also important part of human society and biodiversity. Improving in vitro maturation rate of donkey oocytes can provide the basis for application of ART in donkey, which would ultimately benefit for the expansion of donkey population and conservation of biodiversity and genetic resources (see details in line 360).

Reviewer 2 Report

Li et al evaluated the transcriptomes in oocytes from donkeys at the germinal vesicle (GV) and metaphase II (MII) stages after in vitro maturation. In this study, cumulus-oocyte complexes (COCs) were collected from donkey ovaries at the commercial abattoirs. COCs were then cultured for 34 hours to allow for in vitro maturation. Oocytes at GV or MII were isolated and processed for single-cell RNA-sequencing (scRNA-seq) analysis using the SMART-seq platform. The authors found that the overall transcripts of MII were lower than that of GV. In addition, there were novel genes discovered at the GV stage. There were some overlaps between transcripts in the GV vs. MII oocytes as well as some similarities. Genes involved in N-glycan biosynthesis were enriched in the MII oocytes compared to GV. Although the findings are interesting, there are some concerns associated with the study.

Major concerns:

  1. The authors need to validate the findings from scRNA-seq to confirm the results i.e., qPCR validation or immunofluorescent analysis of oocytes from GV vs. MII stages. This is important especially the genes that the authors found to be unique to GV or those N-Glycan-related genes that were enriched in the MII oocytes. This is crucial and would provide the rigor for this study.
  2. The authors did not include the control group where MII oocytes are collected from the oviducts to represent in vivo matured oocytes. It is possible that these changes in transcriptional profile were due to the artifact during in vitro culture. This needs to be included as the experimental control group. If it is unlikely that this group can be included, the authors need to add the discussion regarding this aspect as the limitation of the study.
  3. The authors mentioned in the abstract and the introduction that the findings from this study could be useful for donkey production. However, the authors did not provide any context/reference/citation regarding the current limitation or drawback about the donkey in vitro maturation process or the IVF in donkeys in general. This could have increased the justification of the study.
  4. Similar to the comments above, the authors need to provide citations/references regarding the statement in lines 42-44. Are donkeys truly under extinction threat? The authors need to provide the source of this statement.
  5. The title of the study is misleading. The authors stated, “Single-Cell RNA-seq Revealed Candidate Genes Affect Donkey Oocytes in vitro Maturation”. The authors used the word “affect”. If that is the case, the authors need to pinpoint the gene, provide the evidence/data to support the statement, ie. Use of the gene knockdown to demonstrate that a lack of certain genes ‘affects’ donkey oocyte in vitro maturation. But the authors have not provided any data or perform experiments to support the statement.

Author Response

Thank you very much for your suggestions and the comments, we have revised our manuscripts point by point according to these comments and suggestions. We wish that our revised manuscript will be qualified enough to be accepted. If there is any questions, please don't hesitate to contact me.

  1. The authors need to validate the findings from scRNA-seq to confirm the results i.e., qPCR validation or immunofluorescent analysis of oocytes from GV vs. MII stages. This is important especially the genes that the authors found to be unique to GV or those N-Glycan-related genes that were enriched in the MII oocytes. This is crucial and would provide the rigor for this study.

Reply: Thanks for your suggestion. To validate the findings from scRNA-seq in this study, we expression of 4 differentially expressed genes were tested using qRT-PCR. We selected two maternal genes (GDF9 and BMP15), a most significantly differential expressed genes, LGALS3, and a N-Glycan-related gene (ALG5). Results showed that the expression of these genes using qRT-PCR is consisted with the RNA-seq data (see in Figure 5). We feel sorry that we cannot test more genes as the stored samples were all run out, and new samples are currently not available. We have supplemented the qRT‐PCR test in the paper (please see in line 142).

  1. The authors did not include the control group where MII oocytes are collected from the oviducts to represent in vivo matured oocytes. It is possible that these changes in transcriptional profile were due to the artifact during in vitro culture. This needs to be included as the experimental control group. If it is unlikely that this group can be included, the authors need to add the discussion regarding this aspect as the limitation of the study.

Reply: Thanks for your question. In fact, we originally planned to use the in vivo matured oocytes as a control group, however, we didn’t get the farmer’s consent for the donkey embryos are too precious. In this study, we select the best quality MII oocytes for RNA-seq as much as possible, according to the embryo morphology evaluation. Considering that these oocytes can develop into embryos just as the in vivo matured oocytes do, we believe that their transcripts are similar to a certain extent. Besides, our study mainly focused on gene expression patterns of in vitro matured donkey oocytes, although artifact during in vitro culture may change the transcriptional profiles to some extent comparing the in vivo maturated oocytes, the data and analysis about in vitro matured donkey oocytes are representable. We have supplemented the above discussion in the paper (please see in line 351).

  1. The authors mentioned in the abstract and the introduction that the findings from this study could be useful for donkey production. However, the authors did not provide any context/reference/citation regarding the current limitation or drawback about the donkey in vitro maturation process or the IVF in donkeys in general. This could have increased the justification of the study.

Reply: Thank you for your suggestions. We have supplemented reference researches about the current application and limitation of in vitro maturation of donkey oocytes in the paper (please see in line 52).

  1. Similar to the comments above, the authors need to provide citations/references regarding the statement in lines 42-44. Are donkeys truly under extinction threat? The authors need to provide the source of this statement.

Reply: Thank you for question. Donkey was majorly contributed to human mobilization to long-distance, loading dramatically enhanced human mobility and were also engaged in promoted the development of agriculture in the human history. While nowadays, donkeys were ignored for the rapid development of industrialization and mechanization, and the donkey population is significantly decreased since the last century. In addition, donkey is in the Red List of endangered animal species of the International Union for the Conservation of Nature. (Ref. Goudet, G., Douet, C., Kaabouba-Escurier, A., et al. Establishment of conditions for ovum pick up and IVM of jennies oocytes toward the setting up of efficient IVF and in vitro embryos culture procedures in donkey (Equus asinus). Theriogenology. 2016. 86(2): p. 528-35; Abdoon, A.S., Fathalla, S.I., Shawky, S.M., et al. In Vitro Maturation and Fertilization of Donkey Oocytes. Journal of Equine Veterinary Science. 2018. 65: p. 118-122.). Although we did not study on endangered donkey species, our results can provide a reference for the conservation of the endangered species. The above information was also added in the paper (please see in line 35).

  1. The title of the study is misleading. The authors stated, “Single-Cell RNA-seq Revealed Candidate Genes Affect Donkey Oocytes In vitro Maturation”. The authors used the word “affect”. If that is the case, the authors need to pinpoint the gene, provide the evidence/data to support the statement, ie. Use of the gene knockdown to demonstrate that a lack of certain genes ‘affects’ donkey oocyte in vitro maturation. But the authors have not provided any data or perform experiments to support the statement.

Reply: Thank you for your question. In this study, we aimed to identify the candidate genes that might influence in vitro maturation of donkey oocytes, rather than focusing on the effect of specific genes. To make the title more accurate, we have changed it to “Single-cell RNA-seq revealed gene expression pattern during in vitro maturation of donkey oocytes” (please see in line 2).

Round 2

Reviewer 1 Report

Thank you for revised version of the manuscript , which is now much better

Reviewer 2 Report

The authors have addressed all of my prior concerns.